# OpenReview forum: "Differentially Private Synthetic Data via APIs 3: Using Simulators Instead of Foundation Models"
_TMLR — Under review for TMLR_

### Review · Reviewer_7hE2 · 2026-04-13

**Summary Of Contributions:**

This paper extends Private Evolution (PE) by incorporating simulators into its framework, aiming to address the performance degradation caused by distribution shift between private data and foundation model pretraining data. The motivation is well-founded, particularly for domains where high-quality simulators are more readily available than well-aligned foundation models. Additionally, the proposed hybrid approach that combines simulators with foundation models is promising, with empirical evidence suggesting that such integration can outperform either component alone.

There are also some weaknesses and questions, which will be shown below.

**Audience:**

Yes

**Audience Explanation:**

This finding maybe useful in this field, but it does NOT reach SOTA

**Broader Impact Concerns:**

No.

**Claims And Evidence:**

Yes

**Claims Explanation:**

There are experimental results

**Requested Changes:**

Weaknesses
1. The privacy guarantees and convergence properties are inherited directly from prior PE work. While this is acceptable, it means the contribution is primarily empirical and system-level rather than theoretical.
﻿
2. The result could not reach SOTA in experiments. What's the contribution of this Sim-PE, since we can directly use the SOTA method.
﻿
Questions
1. For Sim-PE, which part use simulator, and witch part use PE?
It seems that RANDOM-API does not use any one of them, although its name has "API", and VARIATION-API use only the simulator (not APL),  although its name has "API".
2. For algorithm, from the main context, I could get to know how many examples generated in each itertaion,  how many examples  in first random-sample stage of each iteration.
3. In P6, to define nearest neighbors by closeness, how the calculate the distance and  clossess for categorical parameters?
4. Sec 4.2, what are the generated data? Policy or distribution or others?
5. How are categorical vs. numerical parameters handled differently in VARIATION_API in practice? Are there heuristics or tuning parameters that significantly affect performance?
6. The method is motivated by scenarios where simulators are naturally available (e.g., networking, physical systems), Can the authors provide results or case studies in the other domains, where simulators are more realistic and commonly used?

---

> ### Author Response · Authors · 2026-04-28
> **Rebuttal (1/3)**
>
> Thank you very much for recognizing that the motivation is well-founded and that the proposed hybrid approach combining simulators and models is promising. We also sincerely appreciate your thoughtful and constructive feedback, which has been very helpful in improving the paper.
>
> Below, we provide point-by-point responses to your questions. We have also updated the manuscript accordingly; all changes are highlighted in red.
>
> ---
>
> ## **Q1. Clarifications on the algorithm and APIs.**
>
> Thank you for the questions. We provide clarifications below.
>
> **(1) Algorithm pipeline.** (Re: “For Sim-PE, which part use simulator, and witch part use PE?”)
>
> PE and simulators are not separate components. Rather, PE defines the overall workflow, and it invokes simulators (through the RANDOM_API and VARIATION_API) as subroutines. In our submission, we include the full pseudocode in Appendix A and a summary in Section 3.1. Specifically, PE (or Sim-PE) first uses **RANDOM_API** to generate an initial set of random samples. It then iteratively refines these samples using the private data:
>
> - Each private sample casts a vote for its closest synthetic sample, yielding a histogram (**DP_NN_HISTOGRAM**) that reflects alignment with private data. Gaussian noise is added to ensure differential privacy.
> - We sample synthetic data according to the noisy histogram, giving higher probability to better-aligned samples.
> - We apply **VARIATION_API** to generate new variants, which serve as input to the next iteration.
>
> The final iteration produces the DP synthetic dataset.
>
> If space permits, we will move the pseudocode from the appendix to the main text to improve clarity.
>
> ---
>
> **(2) How RANDOM_API and VARIATION_API work.** (Re: “It seems that RANDOM-API does not use any one of them”)
>
> Both APIs use simulators to generate samples (Section 3.2). Specifically, RANDOM_API samples simulator parameters at random, while VARIATION_API perturbs the parameters of an existing sample and then re-invokes the simulator to generate a new sample.
>
> ---
>
> **(3) Naming of the APIs.** (Re: “VARIATION-API use only the simulator (not APL), although its name has "API".”)
>
> “RANDOM_API” and “VARIATION_API” are terms introduced in the original Private Evolution paper (Lin et al., 2023). They refer to abstract interfaces for generating random and variation samples within the PE framework. Prior work implements these APIs using foundation models; in Sim-PE, we instead implement them using simulators. We keep the terminology for consistency with prior work.
>
> ---
>
> ## **Q2. The number of generated examples.**
>
> The number of samples generated in the initial stage and in each iteration is the same, denoted by $N_{\text{syn}}$ in the pseudocode (Algorithm 1 in Appendix A). In our experiments (Section 4.1), we follow the DPImageBench setting (Gong et al., 2025) and set $N_{\text{syn}} = 60{,}000$ (Appendix D.1–D.3).
>
> If space permits, we will move the pseudocode from the appendix to the main text to improve clarity.
>
> ---
>
> ## **Q3. Calculating distance for categorical parameters.**
>
> We clarify that distances are not computed on the parameters themselves (whether numerical or categorical), but on the **generated samples**. We have added the following clarification in Section 3.2:
>
> > A simulator generates samples based on a set of input parameters. These parameters are not part of the generated samples themselves; rather, they act as control signals that determine certain properties of the outputs.
>
> Thank you for pointing this out.
>
> ---
>
> ## **Q4. The generated data in Section 4.2.**
>
> Thank you for the question.
>
> In this setting, RANDOM_API and VARIATION_API operate on four numerical parameters that define a policy. These parameters map observations (cart position, cart velocity, pole angle, and pole angular velocity) to control signals (moving the cart left or right) via a linear function. Executing these control signals in the simulator produces the final trajectories.
>
> These details were originally included in Appendix D.4. Following your suggestion, we have added a “Sim-PE adoption” paragraph in Section 4.2 to improve clarity.

---

> > ### Author Response · Authors · 2026-04-28
> > **Rebuttal (2/3)**
> >
> > ---
> >
> > ## **Q5. Categorical and numerical parameter tuning.**
> >
> > In our submission, we provide sensitivity analyses on: (1) the ranges of simulator parameters, and (2) the Sim-PE hyperparameters controlling variation degrees. We summarize the key results from Appendices F.5 and F.3 as below:
> >
> > **(1) Parameter ranges (Appendix F.5).**
> > We study how prior knowledge about parameter ranges affects performance. Regardless of the specific ranges, Sim-PE consistently improves data quality over iterations and significantly outperforms standard PE.
> >
> > **(2) Variation hyperparameters (Appendix F.3).**
> > Following the PE paper (Lin et al., 2023), we gradually decrease the variation degrees across iterations. Larger variations in early iterations encourage exploration, while smaller variations later support convergence. Further tuning of these parameters may yield additional improvements.
> >
> >
> >
> > ## **Q6. Results in other domains.**
> >
> > > If we understand correctly, the reviewer is asking for results beyond *image* domains, particularly in settings where simulators are naturally available (e.g., networking, physical systems). If this interpretation is incorrect, we would be happy to clarify further or provide additional experiments.
> >
> > Our submission already includes a non-vision task in **physical systems** (specifically, robotics simulation) (Sections 4.2 and 5.4). To further address your concern, we have additionally included a new experiment in the **networking** domain (added to Appendix E.3 in the revision).
> >
> > Below, we introduce these two domains in turn.
> >
> >
> > ### (1) Robotics simulator
> >
> > **The setting.** We consider a new setting where the goal is *not* to generate observations directly, but to generate **DP control signals** (e.g., policies) whose execution in a simulator reproduces the distribution of private observations (Section 3.4).
> >
> > For example,  In the CartPole environment (Section 4.2), the private data consist of trajectories (e.g., sequences of states such as position and velocity). However, for downstream applications, the goal is to generate **control policies** (i.e., actions that move the cart left or right). Thus, instead of generating trajectories directly, we aim to generate policies such that, when executed in the simulator, they produce trajectories that match the private data distribution.
> >
> > **Why prior methods do not apply.** Existing DP methods are not suitable for this setting for two fundamental reasons (Section 3.4):
> > 1. **Different data spaces.** Prior methods assume the private data and generated data lie in the same space (e.g., image → image). Here, the private data are *observations* (trajectories), while the generation target is *latent control signals*, which are not directly observed.
> > 2. **Non-differentiable pipeline.** A natural extension would be to generate control signals, run them in the simulator, and match outputs to private data. However, this requires backpropagation through the simulator, which is typically **non-differentiable**, making gradient-based DP training infeasible.
> >
> > **Why Sim-PE works.** Sim-PE naturally handles this setting by treating the simulator as the generation mechanism and directly optimizing over its input parameters (control signals) using the PE framework, without requiring gradients or matching data spaces.
> >
> > **Experimental setup.** We use the CartPole environment (Section 4.2), where policies are parameterized as simple linear functions mapping observations to actions. We evaluate:
> > - **Utility:** whether generated policies succeed in balancing the pole (classification accuracy).
> > - **Fidelity:** Wasserstein distance between trajectory length distributions.
> >
> > **Key results (Section 5.4).**
> > - Sim-PE steadily improves both utility and fidelity over iterations (e.g., accuracy increases from ~50% to 76%, and Wasserstein distance decreases by ~4×).
> > - Performance further improves with larger privacy budgets (reaching up to 96% accuracy at higher $\epsilon$).
> >
> > These results demonstrate that Sim-PE enables fundamentally new DP applications that are not directly supported by prior methods.
> >
> > Please refer to Sections 4.2, 5.4, and Appendix D.4 for more details and results.

---

> > > ### Author Response · Authors · 2026-04-28
> > > **Rebuttal (3/3)**
> > >
> > > ### (2) Networking data
> > >
> > >
> > > We follow the experimental setup of NetDPSyn [1], a recent paper published at IMC 2024, one of the top networking conferences.
> > >
> > > **Data.**  We aim to generate differentially private *network flow header* data using the TON dataset [1] as the private dataset. Each sample contains the following fields:
> > > - Source IP address
> > > - Destination IP address
> > > - Source port
> > > - Destination port
> > > - Protocol (e.g., UDP, TCP)
> > > - Timestamp of the first packet
> > > - Duration of the flow
> > > - Number of packets
> > > - Number of bytes
> > > - Flow label (10 classes, e.g., normal, background, DDoS)
> > >
> > > **RANDOM and VARIATION APIs.**  We follow the same method as in Section 3.2.1. Specifically, the `RANDOM_API` generates random flows, and the `VARIATION_API` perturbs individual fields of a flow (e.g., modifying the IP address).
> > >
> > > **Embedding.**  To keep the setup minimal, we embed samples directly in their normalized raw format (one-hot encodings for categorical fields and normalized numeric values for continuous ones). Thus, **the embedding does not rely on any machine learning model**.
> > >
> > > **Metrics.**  Following [1], we report:
> > > - **1-way and 2-way Total Variation Distance (TVD)** between synthetic and private data (lower is better)
> > > - **Classification accuracy**, obtained by training a classifier on synthetic data and testing on held-out private data (higher is better). This represents the downstream task of network flow data.
> > >
> > > **Privacy budget.**  As in [1], we set ε = 2.
> > >
> > > **Key results.**  The table below shows how the metrics evolve across Sim-PE iterations. Key observations:
> > >
> > > 1. **All metrics steadily improve across iterations.** Final performance is significantly better than the initial random samples (e.g., accuracy improves by **16.76%**). This demonstrates that Sim-PE can work effectively for networking data *even without* ML-based embeddings or a sophisticated simulator.
> > >
> > > 2. **Compared to Figure 3 in [1], this simple approach already outperforms two baselines in accuracy.** Notably, NetShare—a widely used DP fine-tuning–based synthesizer in networking (160+ citations)—is surpassed by Sim-PE in classification accuracy. This highlights the strong potential of Sim-PE beyond the image domain.
> > >
> > > We have added these experiments to Appendix E.3 in the revision.
> > >
> > > | Iteration | Accuracy | 1-way TVD | 2-way TVD |
> > > |-----------|----------|-----------|-----------|
> > > | 0         | 56.48    | 0.29      | 0.46      |
> > > | 1         | 56.48    | 0.29      | 0.45      |
> > > | 2         | 56.48    | 0.28      | 0.44      |
> > > | 3         | 56.75    | 0.27      | 0.43      |
> > > | 4         | 57.08    | 0.27      | 0.42      |
> > > | 5         | 58.28    | 0.26      | 0.41      |
> > > | 6         | 60.29    | 0.25      | 0.40      |
> > > | 7         | 61.28    | 0.24      | 0.38      |
> > > | 8         | 63.47    | 0.23      | 0.37      |
> > > | 9         | 65.95    | 0.22      | 0.36      |
> > > | 10        | 67.37    | 0.21      | 0.35      |
> > > | 11        | 68.47    | 0.21      | 0.33      |
> > > | 12        | 72.18    | 0.20      | 0.32      |
> > > | 13        | 72.77    | 0.19      | 0.31      |
> > > | 14        | 73.24    | 0.18      | 0.30      |
> > >
> > > [1] https://arxiv.org/abs/2409.05249
> > >
> > >
> > >
> > > ## **Weaknesses**
> > >
> > > Thank you for raising these points. We would like to highlight our key contributions (as also recognized and emphasized by reviewer SYyw):
> > >
> > > (1) Sim-PE addresses two important practical limitations of PE: (a) poor performance under distribution shift, and (b) high cost due to repeated foundation-model API calls.
> > >
> > > (2) Sim-PE enables new applications where prior approaches are not directly applicable; see Q6.
> > >
> > > We have revised the contribution paragraph in Section 1 to better highlight these points.
> > >
> > > ---
> > >
> > > Thank you again for your valuable feedback and for helping us improve the paper. If you have any additional questions, please feel free to post them here—we will be available throughout the discussion phase.

---

### Review · Reviewer_uXoz · 2026-04-14

**Summary Of Contributions:**

This paper presents a way to simulate DP images without using LLM or foundational models.

**Additional Comments:**

N/A

**Audience:**

Yes

**Audience Explanation:**

Generating DP data in domains where data privacy is important, but foundation models are lacking, is an important application of ML/AI that many of TMLR's audience would find relevant

**Claims And Evidence:**

Yes

**Claims Explanation:**

Empirical results are reasonable. However, MNIST and CIFAR are very old image data sets and they are less relevant today. See "requested changes" below.

**Requested Changes:**

My main request for change is to enhance the experiments.

1. MNIST and CIFAR are very old image data sets and are not very relevant now. If the authors do want to focus exclusively on vision tasks, is it possible to use some modern data sets? Or at least data sets that are larger, richer, and more challenging.

2. Do the authors consider extending their framework to non-vision tasks, such as NLP or LLMs? The task considered here would be relevant for LLM pretraining as de-sensitivized data could be generated very fast, without consuming tokens.

---

> ### Author Response · Authors · 2026-04-28
> **Rebuttal (1/2)**
>
> Thank you very much for recognizing the results of the paper and its relevance to the TMLR community. We also sincerely appreciate your thoughtful and constructive feedback, which has been very helpful in improving the paper.
>
> Below, we provide point-by-point responses to your questions. We have also updated the manuscript accordingly; all changes are highlighted in red.
>
> ---
>
> ## **Q1. Datasets other than MNIST and CIFAR.**
>
> Thank you for the question.
>
> First, we would like to clarify that, in addition to MNIST and CIFAR10, we also use CelebA as a dataset (Section 4.1.1). We show that Sim-PE improves PE’s performance on CelebA (Table 1), and that combining simulators and foundation models outperforms using either one alone on this dataset (Table 2). Additional results and analysis on CelebA can be found in Section 5 and Appendix F.
>
> Second, and more importantly, our work does *not* focus exclusively on vision tasks. In our submission, we introduce a new application involving a robotics simulator, where existing DP fine-tuning approaches are not directly applicable. We believe this application highlights the unique value of Sim-PE. In addition, to further address your question, we include new experiments on synthesizing networking data. Please see Q2 for more details on these two tasks.
>
> ---
>
> ## **Q2. Non-vision tasks.**
>
> Thank you for the question. Our submission already includes a non-vision task. To further address your concern, we have additionally included a new experiment in another non-vision domain.
>
> Specifically, in the following, we will first describe the robotics simulator experiments included in the submission (Sections 4.2 and 5.4). In addition, we introduce new experiments on generating networking data, which have been added to Appendix E.3 in the revision.
>
> Regarding NLP (as mentioned by the reviewer), while it is certainly a relevant domain, strong foundation models already exist, and prior work (Aug-PE; Xie et al., 2024) has demonstrated that PE with foundation models can already perform well on NLP tasks. Therefore, we believe exploring Sim-PE in NLP may provide less additional value compared to the domains we study here. We consider this an interesting direction for future work.
>
>
> ### (1) Robotics simulator
>
> **The setting.** We consider a new setting where the goal is *not* to generate observations directly, but to generate **DP control signals** (e.g., policies) whose execution in a simulator reproduces the distribution of private observations (Section 3.4).
>
> For example,  In the CartPole environment (Section 4.2), the private data consist of trajectories (e.g., sequences of states such as position and velocity). However, for downstream applications, the goal is to generate **control policies** (i.e., actions that move the cart left or right). Thus, instead of generating trajectories directly, we aim to generate policies such that, when executed in the simulator, they produce trajectories that match the private data distribution.
>
> **Why prior methods do not apply.** Existing DP methods are not suitable for this setting for two fundamental reasons (Section 3.4):
> 1. **Different data spaces.** Prior methods assume the private data and generated data lie in the same space (e.g., image → image). Here, the private data are *observations* (trajectories), while the generation target is *latent control signals*, which are not directly observed.
> 2. **Non-differentiable pipeline.** A natural extension would be to generate control signals, run them in the simulator, and match outputs to private data. However, this requires backpropagation through the simulator, which is typically **non-differentiable**, making gradient-based DP training infeasible.
>
> **Why Sim-PE works.** Sim-PE naturally handles this setting by treating the simulator as the generation mechanism and directly optimizing over its input parameters (control signals) using the PE framework, without requiring gradients or matching data spaces.
>
> **Experimental setup.** We use the CartPole environment (Section 4.2), where policies are parameterized as simple linear functions mapping observations to actions. We evaluate:
> - **Utility:** whether generated policies succeed in balancing the pole (classification accuracy).
> - **Fidelity:** Wasserstein distance between trajectory length distributions.
>
> **Key results (Section 5.4).**
> - Sim-PE steadily improves both utility and fidelity over iterations (e.g., accuracy increases from ~50% to 76%, and Wasserstein distance decreases by ~4×).
> - Performance further improves with larger privacy budgets (reaching up to 96% accuracy at higher $\epsilon$).
>
> These results demonstrate that Sim-PE enables fundamentally new DP applications that are not directly supported by prior methods.
>
> Please refer to Sections 4.2, 5.4, and Appendix D.4 for more details and results.

---

> > ### Author Response · Authors · 2026-04-28
> > **Rebuttal (2/2)**
> >
> > ### (2) Networking data
> >
> >
> > We follow the experimental setup of NetDPSyn [1], a recent paper published at IMC 2024, one of the top networking conferences.
> >
> > **Data.**  We aim to generate differentially private *network flow header* data using the TON dataset [1] as the private dataset. Each sample contains the following fields:
> > - Source IP address
> > - Destination IP address
> > - Source port
> > - Destination port
> > - Protocol (e.g., UDP, TCP)
> > - Timestamp of the first packet
> > - Duration of the flow
> > - Number of packets
> > - Number of bytes
> > - Flow label (10 classes, e.g., normal, background, DDoS)
> >
> > **RANDOM and VARIATION APIs.**  We follow the same method as in Section 3.2.1. Specifically, the `RANDOM_API` generates random flows, and the `VARIATION_API` perturbs individual fields of a flow (e.g., modifying the IP address).
> >
> > **Embedding.**  To keep the setup minimal, we embed samples directly in their normalized raw format (one-hot encodings for categorical fields and normalized numeric values for continuous ones). Thus, **the embedding does not rely on any machine learning model**.
> >
> > **Metrics.**  Following [1], we report:
> > - **1-way and 2-way Total Variation Distance (TVD)** between synthetic and private data (lower is better)
> > - **Classification accuracy**, obtained by training a classifier on synthetic data and testing on held-out private data (higher is better). This represents the downstream task of network flow data.
> >
> > **Privacy budget.**  As in [1], we set ε = 2.
> >
> > **Key results.**  The table below shows how the metrics evolve across Sim-PE iterations. Key observations:
> >
> > 1. **All metrics steadily improve across iterations.** Final performance is significantly better than the initial random samples (e.g., accuracy improves by **16.76%**). This demonstrates that Sim-PE can work effectively for networking data *even without* ML-based embeddings or a sophisticated simulator.
> >
> > 2. **Compared to Figure 3 in [1], this simple approach already outperforms two baselines in accuracy.** Notably, NetShare—a widely used DP fine-tuning–based synthesizer in networking (160+ citations)—is surpassed by Sim-PE in classification accuracy. This highlights the strong potential of Sim-PE beyond the image domain.
> >
> > We have added these experiments to Appendix E.3 in the revision.
> >
> > | Iteration | Accuracy | 1-way TVD | 2-way TVD |
> > |-----------|----------|-----------|-----------|
> > | 0         | 56.48    | 0.29      | 0.46      |
> > | 1         | 56.48    | 0.29      | 0.45      |
> > | 2         | 56.48    | 0.28      | 0.44      |
> > | 3         | 56.75    | 0.27      | 0.43      |
> > | 4         | 57.08    | 0.27      | 0.42      |
> > | 5         | 58.28    | 0.26      | 0.41      |
> > | 6         | 60.29    | 0.25      | 0.40      |
> > | 7         | 61.28    | 0.24      | 0.38      |
> > | 8         | 63.47    | 0.23      | 0.37      |
> > | 9         | 65.95    | 0.22      | 0.36      |
> > | 10        | 67.37    | 0.21      | 0.35      |
> > | 11        | 68.47    | 0.21      | 0.33      |
> > | 12        | 72.18    | 0.20      | 0.32      |
> > | 13        | 72.77    | 0.19      | 0.31      |
> > | 14        | 73.24    | 0.18      | 0.30      |
> >
> > [1] https://arxiv.org/abs/2409.05249
> >
> > ---
> > Thank you again for your valuable feedback and for helping us improve the paper. If you have any additional questions, please feel free to post them here—we will be available throughout the discussion phase.

---

### Review · Reviewer_SYyw · 2026-04-21

**Summary Of Contributions:**

This paper proposes Sim-PE, an extension of Private Evolution (PE) that allows PE to use simulator APIs in addition to foundation model APIs. The motivation is clear: prior versions of PE often perform poorly when the private data distribution is far from the data mixture distribution of foundational models' prior training stages. The paper’s main contribution is therefore a useful generalization of the PE framework, together with concrete instantiations and experiments across multiple settings. Empirically, the results show that Sim-PE can substantially improve over Aug-PE in domains with strong distribution shift, and that simulators and foundation models can also be combined within a unified pipeline. In particular, the CartPole example is especially compelling, which common DP-finetune based methods do not naturally handle.

Overall, while the novelty is somewhat incremental in the sense that the method remains within the PE framework, I think it is a valuable contribution as it addresses two important practical limitations of prior PE: 1. poor performance under distribution shift and 2. high cost from repeated foundation-model API calls. The paper is well motivated and the experimental tasks are well chosen and thus I recommend for acceptance

**Additional Comments:**

- Some readers may feel that this is a relatively direct instantiation of prior PEs, with simulator APIs replacing foundation-model APIs. I partially agree, but I think the paper still has practical value. The cost of repeated foundation-model API calls in PE can be substantial, especially when generating large scale synth corpus, and PE can degrade badly when the private data is far from the model’s pretraining distribution. Simulators can partially address both issues in domains where an appropriate simulator exists. In that sense, a simulator can act as a domain-specific generator that is better aligned with the private data than a generic foundation model.

- The CartPole task is a strong motivating example. It cleanly illustrates a setting where standard DP-ft based methods or synthesis methods are not naturally applicable.

- The simulator + foundation-model hybrid is intuitive and empirically promising. It would be interesting to study more principled ways of combining the two, since that might further reduce the number of PE iterations or improve sample quality [1]. This could be an interesting future work.

[1] MAPLE: Metadata Augmented Private Language Evolution

**Audience:**

Yes

**Audience Explanation:**

Yes. Privacy-preserving ML and DPsynth data are active areas of interest, and this paper broadens the design space for DP synthesis in a practically meaningful way.

**Claims And Evidence:**

Yes

**Claims Explanation:**

Yes.

**Requested Changes:**

- Please clarify the abstraction in Section 3.2.1: the distinction between numerical and categorical simulator parameters is reasonable, but it is somewhat abstract in the main text. Even though more details appear in the appendix, it would help to include one or two concrete examples directly in the main body. In particular, it would be useful to clarify that these parameters are not necessarily tied to quality, but rather describe controllable attributes of the simulator output depending on the simulator.

- Please sharpen the presentation of the contribution relative to prior PEs. As written, some readers may come away with the impression that this is simply PE with simulator APIs swapped in for foundation-model APIs. I think the paper does make a meaningful contribution, but it would benefit from a more explicit emphasis on the how Sim-PE addresses the weakness of prior PEs and the broader applicability beyond simple text/image synthesis

---

> ### Author Response · Authors · 2026-04-28
> **Rebuttal (1/2)**
>
> Thank you very much for recognizing that Sim-PE provides a valuable contribution in addressing important limitations of PE, that the paper is well motivated, and that the experiments are well chosen. We also sincerely appreciate your thoughtful and constructive feedback, which has been very helpful in improving the paper.
>
> Below, we provide point-by-point responses to your questions. We have also updated the manuscript accordingly; all changes are highlighted in red.
>
> ---
>
> ## **Q1. Clarifications on numerical and categorical parameters.**
>
> Thank you for these insightful comments. We have revised the paper to provide clearer explanations of these parameters in the main text. Specifically, we made the following updates:
>
> 1. We clarify that these parameters are not part of the generated samples, but instead serve as control signals for the simulators.
> 2. We expand the concrete example with additional details to improve clarity.
>
> The updated text (first three paragraphs of Section 3.2.1) is shown below, with changes in bold:
>
> > While different simulators have very different programming interfaces, most of them can be abstracted in the same way. **A simulator generates samples based on a set of input *parameters*. These parameters are *not* part of the generated samples themselves; rather, they act as *control signals* that determine certain properties of the outputs.** **For example, face image renderers in Wood et al. (2021); Bae et al. (2023) generate images conditioned on parameters such as facial pose, lighting intensity, and the IDs of the underlying 3D face models and hairstyles.**
>
> > **These parameters can be further categorized into two types, depending on whether their values admit a natural ordering.** *Numerical parameters* refer to those with meaningful ordering (e.g., 1 is closer to 2 than to 3), whether discrete (e.g., $\in \{0,1,2\}$) or continuous (e.g., $\in [0,2]$). In contrast, *categorical parameters* are those *without* inherent ordering, or where even a small change in the parameter can cause a large, unrelated change in the output. **In the face renderer example above, facial pose and lighting intensity are numerical parameters, as their magnitudes are meaningful and variations lead to smooth changes in the generated images. In contrast, the IDs of the 3D face models and hairstyles are categorical parameters, since different IDs correspond to distinct entities without any notion of proximity or ordering.**
>
> > **Under this formulation, a simulator $\mathcal{S}$ generates a sample denoted by $\mathcal{S}(\xi_1,\ldots,\xi_p,\phi_1,\ldots,\phi_q)$, where $\xi_1,\ldots,\xi_p \in \Xi_1 \times \cdots \times \Xi_p$ are $p$ *categorical* parameters and $\phi_1,\ldots,\phi_q \in \Phi_1 \times \cdots \times \Phi_q$ are $q$ *numerical* parameters.**

---

> > ### Author Response · Authors · 2026-04-28
> > **Rebuttal (2/2)**
> >
> > ---
> >
> > ## **Q2. Presentation of the contribution.**
> >
> > Thank you again for these helpful suggestions. We indeed observed that readers may miss our core contributions, and your feedback highlighted that our presentation of the contributions was not sufficiently explicit or clear. Following your suggestions, we have revised the contribution summary paragraph in Section 1 as follows (changes in bold):
> >
> > > Our key contributions are:
> >
> > > - **Advancing PE.** (1) We discover that PE can leverage tools beyond foundation models and propose Sim-PE, which integrates simulators into PE. **This integration alleviates two key limitations of PE. (a) Improved performance under large distribution shift. PE performs poorly when the private data distribution deviates significantly from the pretraining data of foundation models. By incorporating simulators, Sim-PE achieves substantially better performance in such settings.**  For example, on the MNIST dataset with $\epsilon=1$, downstream classification accuracy improves to 89.1%, compared to 27.9% with the original PE. **(b) Improved efficiency. As an additional benefit, Sim-PE reduces the high cost of repeated foundation model API calls in PE by offloading computation to potentially cheaper simulators. For example, on MNIST, Sim-PE running on CPU is 80$\times$ faster than PE on GPU, while also benefiting from the lower cost of CPU computation.**  (2) We further introduce a unified framework that allows *models and simulators to be used interchangeably*, enabling PE to leverage the complementary strengths of both and yielding additional performance gains.
> >
> > > - Unlocking new DP applications. We show that Sim-PE supports applications in which the generation target and the private observations lie in *different* spaces, such as *generating DP control signals whose execution in a robotics simulator reproduces the distribution of private observations*. This setting is fundamentally challenging for prior training-based methods for two reasons: (1) they are designed for scenarios in which the generation target and observations reside in the same space, and (2) they are gradient-based, making them incompatible with simulators through which backpropagation is not possible.
> >
> >
> > ---
> >
> > ##  **Additional comments.**
> >
> > Thank you for your additional comments recognizing the contributions of Sim-PE in addressing key limitations of PE, the significance of the CartPole task where DP-FT-based methods are not directly applicable, and our promising idea of combining simulators with foundation models. We also agree that exploring better ways to combine the two is an interesting direction for future work. In addition, we have incorporated the relevant work MAPLE into the paper.
> >
> > ---
> >
> > Thank you again for your valuable feedback and for helping us improve the clarity and presentation of our work. If you have any additional questions, please feel free to post them here—we will be available throughout the discussion phase.